# The Effects of Feeding a Whole-in-Shell Peanut-Containing Diet on Layer Performance and the Quality and Chemistry of Eggs Produced †

Kari L. Harding [1], Thien Vu [2], Rebecca Wysocky [1], Ramon Malheiros [1], Kenneth E. Anderson [1] and Ondulla T. Toomer [1,2,*]

[1] Prestage Department of Poultry Science, NC State University, Raleigh, NC 27695, USA; klhardin@ncsu.edu (K.L.H.); becca_wysocky@ncsu.edu (R.W.); rdmalhei@ncsu.edu (R.M.); kanderso@ncsu.edu (K.E.A.)
[2] Food Science & Market Quality and Handling Research Unit, ARS, USDA, Raleigh, NC 27695, USA; Thien.Vu@usda.gov
* Correspondence: Ondulla.Toomer@usda.gov; Tel.: +919-515-9109; Fax: +919-515-7070
† Mention of a trademark or proprietary product does not constitute a guarantee or warranty of the product by the U.S. Department of Agriculture or North Carolina Agricultural Research Service, nor does it imply approval to the exclusion of other products that may be suitable. USDA is an equal opportunity provider and employer.

**Abstract:** The abundance of peanut and poultry production within the state of North Carolina and the US Southeast, led us to conduct a layer feeding trial to determine the utilization of whole-in-shell high-oleic peanuts (WPN) and/or unblanched high-oleic peanuts (HOPN) as an alternative feed ingredient for poultry. To meet this objective, we randomly assigned 576 shaver hens to 4 dietary treatments (4 rep/trt). The dietary treatments consisted of a conventional control diet (C1), a diet containing 4% WPN, an 8% HOPN diet, and a control diet containing soy protein isolate (C2). Feed and water were provided for 6 weeks ad libitum. Pen body weights (BW) were recorded at week 0 and week 6 (wk6), and feed weights were recorded bi-weekly. Shell eggs were collected daily and enumerated. Bi-weekly 120 eggs/treatment were collected for quality assessment and egg weight (EW), while 16 eggs/treatment were collected for chemical analysis. There were no significant differences in BW or EW at week 6. Hens fed the C2 produced more total dozen eggs relative to C1 hens over the feeding trial ($p < 0.05$). Hens fed the C1 diet consumed less total feed relative to the other treatments with the best feed conversion ratio ($p < 0.05$). Most eggs produced from each treatment were USDA grade A, large eggs. There were no differences in egg quality, with the exception of yolk color, with significantly higher yolk color scores in eggs produced from the C1 and C2 treatments relative to the other treatments ($p < 0.05$). Eggs produced from the HOPN treatment had significantly reduced stearic and linoleic fatty acid levels relative to the other treatments ($p < 0.05$). Eggs produced from hens fed the WPN diet had significantly greater β-carotene content relative to eggs from the other treatment groups ($p < 0.05$). In summary, this study suggests that WPN and/or HOPN may be a suitable alternative layer feed ingredient and a dietary means to enrich the eggs produced while not adversely affecting hen performance.

**Keywords:** alternative poultry feed ingredients; high-oleic peanuts; whole-in-shell peanuts; layers; shell eggs

## 1. Introduction

North Carolina is one of the top six peanut producing states (Georgia, Florida, Alabama, Texas, North Carolina, South Carolina), which produce nearly all the US annual peanut crop. [1]. Relating to this, a considerable quantity of peanuts produced do not meet the industry's standards for human consumption. These components include peanut screenings (small sized peanuts), splits (above the 15%), and any peanuts not meeting

the grading or sizing requirements set forth by the Peanut Standards Board and the U.S. Department of Agriculture Marketing Service [2]. Moreover, considerable agricultural waste of peanut shells and skins are produced by the peanut industry during the shelling and blanching process that are discarded due to their little economic value. Peanuts like soybeans are legumes and oilseeds rich in protein, while also providing dietary lipids as energy. Interestingly, peanut skins, the coating of the peanut, provide 15% antioxidant rich polyphenolic compounds, 19% fat, and 12% fiber [3]. Hence, whole unblanched (skin intact) peanuts may serve as a nutrient-rich value-added alternative feed ingredient for animal food production while reducing the production of agricultural waste by-products from the peanut industry. Thus, numerous studies have been conducted to explore the effective utilization of peanuts and/or peanut by-products within the animal feed as value-added feed components [3–10] and to enhance agricultural sustainability.

In addition to peanut production, North Carolina is ranked second in turkey and third in poultry production nationally, with over 5700 family farms producing poultry and eggs with an economic impact of approximately USD 39.76 billion annually [11]. Importing corn and soybean meal, the primary feedstock rations for poultry, from South America [12] and even the U.S. Midwest is costly, but if peanuts and/or peanut by-products are shown to be a competitive alternative, a potential reduction in feed costs without hindering growth and performance of poultry could be gained. Using locally grown peanuts (whole in-shell or unblanched) as feed ingredients to the poultry industry could replace imported corn and soybean meal costs. Peanuts could be a viable alternative to soybeans which may have limited availability due to supply and associated rising cost. As a feed ingredient, peanuts and peanut by-products are good sources of protein, fiber, and fat [13,14]. While a few studies have examined the use of peanut skins [3,5] and fermented peanut shells [15,16] as alternative feed ingredients for poultry, no studies to date have examined the use of whole-in-shell peanuts and few studies have examined the use of unblanched peanuts, as a feed ingredient for poultry.

Aka et al., (2020) demonstrated that fermented peanut shells alone could be substituted into feed in place of rice bran at up to 6% inclusion without affecting bird performance, while another study concluded that fermented peanut shell meal could be included up to 5% of the diet to enhance broiler performance [15,16]. Hence, in this study, we aimed to determine the effect of whole-in-shell peanuts and unblanched peanuts on the performance and production of laying hens. Secondly, we aim to determine the effects of feeding a whole-in-shell peanut diet or unblanched peanut diet on the quality, lipid and fatty acid content of the eggs produced. In our previous layer feeding trials, we demonstrated that layer body weight, and feed intake of hens fed a 24% unblanched peanut diet was similar to that of hens fed a non-conventional control diet containing soy protein isolate, defatted soybean meal, and yellow corn [3]. However, hens fed a 24% unblanched peanut diet produced a significantly fewer total number of eggs compared to hens fed a control diet containing soy protein isolate [3]. Furthermore, eggs produced from hens fed a 24% unblanched peanut diet were significantly smaller in weight compared to eggs produced from hens fed a control diet containing soy protein isolate [3]. Therefore, in this study, we aimed to utilize one-third of the previous inclusion level of peanuts in the diet (8%) of layers to determine the effects on layer production performance and on egg chemistry and quality. We conjecture that hens fed an 8% unblanched peanut diet will have similar egg production and egg weights to hens fed a conventional layer diet (defatted soybean meal + corn) and similar to hens fed a non-conventional control diet containing soy protein isolate. Additionally, we aim to compare the effects of feeding a non-conventional control diet containing soy protein isolate to the effects of feeding a conventional control diet on layer production performance, egg weights, egg quality, and chemistry.

## 2. Materials and Methods

All animal research procedures used in these feeding trials were approved by the North Carolina State University Institutional Animal Care and Use Committee (IACUC #19-761-A).

### 2.1. Experimental Design, Animal Husbandry, and Dietary Treatments

This study was conducted at the North Carolina Department of Agriculture and Consumer Services Piedmont Research Station (Salisbury, NC, USA). Four experimental diets were formulated in Concept 5 (level 2, version 10.0) to be isocaloric (2928 kcal/kg metabolizable energy) and isonitrogenous (19.5% crude protein) with an estimated particle size between 800 and 1000 μm (Table 1). Five hundred seventy-six Shaver laying hens (28 to 34 weeks of age) were randomly assigned to one of the four dietary treatments (144 hens per treatment), with four replicates of thirty-six birds per treatment. Hens were housed with 18 birds per cage allowing a space of 175.26 cm$^2$ per hen. Cages consisted of two rows of a Conventional Tri-Deck Stacked Layer Cage System with 66.04 × 121.92 cm$^2$ (26 × 48 in$^2$) per cage. The study was conducted in a standard height, windowless enclosed ventilated house.

**Table 1.** Composition of formulated experimental laying hen diets.

| | Treatments [1] | | | |
|---|---|---|---|---|
| | Control-1 | HOPN | WPN | Control-2 |
| Feed Ingredients | g/kg DM | | | |
| Corn (yellow) | 518.4 | 518.0 | 512.5 | 542.0 |
| Soybean Meal | 322.4 | 277.9 | 301.6 | 288.3 |
| Calcium Carbonate | 95.5 | 88.9 | 95.0 | 95.8 |
| Dicalcium Phosphate | 18.1 | 26.0 | 18.6 | 18.1 |
| Whole In-Shell Peanut | 0.0 | 0.0 | 40.0 | 0.0 |
| High-Oleic Peanut | 0.0 | 80.0 | 0.0 | 0.0 |
| Sodium Chloride | 2.5 | 2.5 | 2.5 | 2.5 |
| L-Lysine | 0.0 | 0.8 | 1.4 | 0.16 |
| DL-Methionine | 1.8 | 2.0 | 1.9 | 1.7 |
| [2] ADM Soy Protein | 0.0 | 0.0 | 0.0 | 16.0 |
| Soybean Oil | 37.3 | 0.0 | 22.6 | 31.4 |
| [3] Santoquin® | 0.5 | 0.5 | 0.5 | 0.5 |
| Choline Chloride | 0.5 | 0.5 | 0.5 | 0.5 |
| [4] Mineral Premix | 2.0 | 2.0 | 2.0 | 2.0 |
| [5] Vitamin Premix | 0.5 | 0.5 | 0.5 | 0.5 |
| [6] Selenium Premix | 0.5 | 0.5 | 0.5 | 0.5 |
| ME (kcal/kg) | 2928 | 2928 | 2928 | 2928 |

[1] Four experimental isocaloric (2928 kcal/kg) and isonitrogenous (19.5% crude protein) diets were formulated: Control-1 = conventional diet containing defatted soybean meal and corn; HOPN = diet containing defatted soybean meal, corn, and 8% unblanched (skin intact) high oleic peanuts; WPN = diet containing defatted soybean meal, corn and 4% whole in shell high-oleic peanuts; Control-2 = diet containing defatted soybean meal, corn, and soy protein isolate. Aflatoxin-free peanuts were used in the preparation of all peanut-containing diets. DM = dry matter. [2] Soy Protein Isolate = purchased from ADM, Chicago, IL, USA. [3] Santoquin® = Feed antioxidant and preservative to prevent fat oxidation in stored feed (Novus International, St. Charles, MO, USA).[4] Mineral premix provides per kg of diet: manganese, 120 mg; zinc, 120 mg; iron, 80 mg; copper, 10 mg; iodine, 2.5 mg; and cobalt. [5] Vitamin premix provides per kg of diet: 13,200 IU vitamin A, 4000 IU vitamin D3, 33 IU vitamin E, 0.02 mg vitamin $B_{12}$, 0.13 mg biotin, 2 mg menadione (K3), 2 mg thiamine, 6.6 mg riboflavin, 11 mg d-pantothenic acid, 4 mg vitamin $B_6$, 55 mg niacin, and 1.1 mg folic acid. [6] Selenium premix = 1 mg Selenium premix provides 0.2 mg Se (as $Na_2\ SeO_3$) per kg of diet. ME = metabolizable energy.

Throughout the feeding trial, birds were provided 14 L:10 D and feed and water ad libitum for 6-weeks. Pen body and feed weights were recorded bi-weekly. Shell eggs were collected and enumerated daily from each pen and replicated and totaled each week. Total number of eggs produced per replicate for each treatment was calculated for the total 6-week feeding trial. The average feed conversion ratio (FCR) was calculated as total

feed consumed over the 6-week feeding trial (kg)/total dozens of eggs produced for each treatment group over the 6-week feeding trial.

The conventional control diet (control-1) was prepared as a conventional layer diet with solvent extracted defatted soybean meal + yellow corn, while the second non-conventional control diet (control-2) was prepared using soy protein isolate (ADM, Chicago, IL, USA) + solvent extracted defatted soybean meal + yellow corn. The whole-in-shell peanut diet (WPN) was prepared using 4% whole-in-shell high-oleic peanuts + solvent extracted defatted soybean meal + yellow corn, and the unblanched high oleic peanut diet (HOPN) was prepared using 8% unblanched (skin intact) high-oleic peanuts + solvent extracted defatted soybean meal + yellow corn. Aflatoxin-free peanuts were used in all peanut-containing experimental diets.

Whole-in-shell high-oleic peanuts were chemically analyzed by ATC Scientific (Little Rock, AR, USA). The nutritional content for whole-in-shell high-oleic peanuts was determined to be: 34.8% crude fat, 22.0% crude protein, 0.16% calcium, 0.30% phosphorous, 34.2% carbohydrates, <5 ppm β-carotene, gross energy 6500 kcal/kg using standard Association of Official Analytical Chemists (AOAC)-approved methods for nuts and seeds with crude fat determination using Gravimetric methods for nuts-AOAC 948.22, protein determination using Kjeldahl method for nuts-AOAC 950.48, mineral determination by elemental analysis of mineral by atomic absorption spectroscopy, carbohydrates were determined using standard colorimetric assay determination and spectroscopy, enzymatic-gravimetric methods were used for carbohydrate determination-AOAC 991.43, standard bomb calorimetry methods were used to determine gross energy, and β-carotene was determined using standard high-performance liquid chromatography and spectrophotometry methods [17–19]. Whole-in-shell high-oleic peanuts and unblanched high-oleic peanuts were crushed using a Roller Mill to form crumbles, prior to inclusion in the finished diets. Each of the experimental diets was supplemented with vitamin, mineral, and selenium premixes manufactured at the NC State University Feed Mill (Raleigh, NC, USA) to meet and/or exceed poultry requirements for vitamins, minerals, and selenium. All four experimental diets were fed as mash diets and were analyzed by the North Carolina Department of Agriculture and Consumer Services and the Food and Drug Protection Division Laboratory (Raleigh, NC, USA) for aflatoxin and microbiological contaminants. All feed ingredients and feed samples were verified to be free of microbiological contaminants. All experimental diets were analyzed for crude fat, total cholesterol, fatty acid profile, and β-carotene by an AOAC-certified lab, ATC Scientific (Little Rock, AR, USA), using AOAC approved standard methods described by Toomer et al. [6].

### 2.2. Egg Quality and Grading

Egg quality was conducted bi-weekly (0, 2, 4, 6) using a 24 sub-sample of eggs randomly selected from each treatment (6 eggs/replicate) in the Egg Quality Lab, Prestage Department Poultry Science, NC State University (Raleigh, NC, USA). Egg quality parameters measured included shell strength, vitelline membrane elasticity (VME), vitelline membrane hardness (VMH), vitelline membrane work of penetration (VMW), egg weight, albumen height, Haugh unit (HU), yolk color, shell color, and shell thickness. Eggshell strength was determined using a texture analyzer (TA-HDplus) with a 250 kg load cell measuring in grams of force. The TA-HDplus has a trigger force of 0.02 kg and a testing speed of 1 mm/s. Vitelline membrane strength was determined using the TA.XTplus Texture Analyzer (Stable Micro Systems, Surrey, United Kingdom) with a 1mm blunt probe with a 500-g load cell per the manufacturer's instructions. The trigger force was 0.0001 kg with a 3.2 mm/s testing speed. Haugh Unit and albumen height were analyzed using the TSS QCD System (Technical Services and Supplies, Dunnington, York, UK). HU is calculated using the following calculation = 100Log (h − 1.7w + 7.6), with h = egg albumen height and w = weight of egg, with values ranging from 0 to 130 and HU scores below 60 for un-fresh eggs [20]. Yolk color was also determined using the TSS QCD System yolk color scan. Yolk color scan was calibrated using the DSM Yolk Color Fan that determines the color

density from lightest to darkest with a range of 1 to 15 [21]. Shell color was determined using refractometry of black, blue, and red wavelengths combined to provide a score from 83.3% (white) to 0% (black). USDA shell egg grading and sizing were conducted on a 120 sub-sample of eggs randomly selected from each treatment group (30 eggs/replicate) bi-weekly.

### 2.3. β-Carotene, Lipid, and Fatty Acid Analysis

At week 0 and week 6, a total of 128 eggs (64 eggs at each time point) were randomly selected, with 16 eggs per treatment (4 eggs randomly selected per replicate) for lipid content (total cholesterol, crude fat, and fatty acid profile) and β-carotene analysis by ATC Scientific using AOAC approved methods. Each egg sample was mixed for homogeneity in a Whirl-pak® (Millipore Sigma, St. Louis, MO, USA) bag for 30 s in a Smasher™ Lab Blender (Weber Scientific, Hamilton, NJ, USA), the homogenous egg sample was pipetted into a 50 mL conical tube and frozen at −20 °C and stored until analysis within 2 weeks of collection. Frozen homogenous egg samples were shipped on dry ice overnight to the vendor for analysis. Total cholesterol, crude fat, and fatty acid analysis was conducted using direct methylation methods, as described by Toomer et al. [6]. Total cholesterol was measured as mg cholesterol/100 g sample weight (feed or egg), while crude fat was measured as a percentage of gram crude fat/gram sample weight (feed or egg). Fatty acid content was measured as a percentage of gram of fatty acid/gram total lipid content of a sample (feed or egg). Methods used to determine β-carotene content in eggs are detailed in the AOAC 958.05 color of egg yolk method [22]. Egg fat hydrolysis methods were determined using the AOAC method 954.02 [23].

### 2.4. Statistical Analysis

Each replicate served as the experimental unit for all variables (body weights, egg weights, feed intake, total dozens of eggs produced, feed conversion ratio). All performance data was evaluated for significance by one-way analysis of variance (ANOVA) at a significance level of $p < 0.05$ using JMP statistical software (version 15.2.1, SAS, Cary, NC, USA). If ANOVA results were significant ($p < 0.05$), a Tukey's multiple comparisons t-test was conducted to compare the mean of each treatment group with the mean of every other treatment at $p < 0.05$ significance level. The individual egg served as the experimental unit for analysis of all egg quality measurements (120 eggs per treatment, 30 eggs/replicate at each time point) and egg chemistry data (16 eggs per treatment, 4 eggs/replicate at each time point of collection) including crude fat, total cholesterol, fatty acid profile, and β-carotene content between the four treatment groups was conducted for significance by one-way analysis of variance (ANOVA) at a significance level of $p < 0.05$ using JMP statistical software (version 15.2.1, SAS, Cary, NC, USA).

## 3. Results and Discussions

Chemical analysis of the four dietary treatments confirmed that the formulated diets were isocaloric, isonitrogenous, and provided adequate calcium and phosphorous levels for laying hens (Table 2). Dietary palmitic saturated fatty acid, the level was the highest in control-1 treatment group. Palmitoleic acid, an omega-7 monounsaturated fatty acid, levels were highest in the HOPN and control-1 experimental diets, relative to the other treatment groups. In vivo, palmitoleic acid is biosynthesized from palmitic acid and can be found in breast milk, animal fats, and vegetable oils. Palmitoleic acid has been shown to act as an anti-inflammatory, to reduce insulin resistance, and reduce hypertension [24]. As expected, oleic monounsaturated fatty acid content was highest in the HOPN and WPN dietary treatments relative to the control group. β-carotene content was negligible (<5 ppm) in all experimental treatments and was below the threshold level of assay detection (data not shown). Linoleic acid content was highest in both control dietary treatments relative to the other treatment groups. Linolenic acid content was highest in the control groups and lowest in the HOPN dietary treatment, while homo-gamma-linolenic acid (GLA)

levels were similar between all dietary treatments (Table 2). GLA is an omega 6, 18 carbon, polyunsaturated fatty acid found commonly in human breast milk, and in the oils of several botanical seeds and dietary supplementation has been shown to reduce inflammatory biomarkers in several chronic inflammatory diseases, such as rheumatoid arthritis and atopic dermatitis [25]. Unexpectantly, elaidic acid content was the highest in the HOPN diet relative to the other treatment groups.

**Table 2.** Chemical and fatty acid analysis of experimental laying hen control and peanut-containing diets.

| | Treatments [1] | | | |
| | Control-1 | HOPN | WPN | Control-2 |
| Nutrient | g/kg DM | | | |
|---|---|---|---|---|
| Crude fat [2] | 56.4 | 80.7 | 65.7 | 52.2 |
| Calcium | 28.2 | 33.4 | 35.0 | 33.8 |
| Phosphorus | 6.30 | 7.60 | 6.80 | 6.70 |
| Palmitic (16:0) * | 123 | 95.0 | 95.6 | 109 |
| Palmitoleic (16:1) * | 7.50 | 6.90 | 1.60 | 3.50 |
| Stearic (18:0) * | 43.4 | 30.7 | 34.90 | 38.50 |
| Oleic (18:1) * | 230 | 593 | 394.0 | 219.0 |
| Elaidic (C18:1 trans) * | 0.50 | 0.80 | 0.40 | 0.90 |
| Linoleic (18:2) * | 510 | 198 | 390 | 533 |
| Linolenic (18:3) * | 62.1 | 12.8 | 44.6 | 63.4 |
| Total Omega 3 | 63.2 | 15.1 | 46.1 | 65.3 |
| Homo-ɤ-linolenic (18:3n − 6) * | 0.40 | 0.70 | 0.60 | 0.60 |

[1] Dietary treatments: Control-1 = conventional diet containing defatted soybean meal and corn; HOPN = diet containing defatted soybean meal, corn, and 8% unblanched (skin intact) high oleic peanuts; WPN = diet containing defatted soybean meal, corn, and 4% whole in shell high-oleic peanuts; Control-2 = diet containing defatted soybean meal, corn, and soy protein isolate. Four dietary treatments were chemically analyzed by AOAC-certified lab, (ATC Scientific, Little Rock, AR, USA) using standard AOAC-approved methods. [2] Crude Fat content = g crude fat/g total sample weight * 100, * Fatty acid content = g of fatty acid/g total lipid content * 100.

There were no mortalities or health related issues associated with any of the treatment hens during the 6-week feeding trial. There were no significant differences in the variables measured between treatment replicates. Moreover, there were no significant treatment differences in the average pen body weights (Table 3) over the 6-week feeding trial. This is contrary to our previous layer feeding trials, which showed hens fed a 24% or 20% HOPN diet had reduced body weights and egg weights relative to the study controls [3,10], suggesting that higher inclusion levels of HOPN may significantly alter body and egg weights in egg-producing layers. In this study, there were no significant differences in average egg weights between the treatment groups over the course of this study, suggesting that the inclusion of 8% HOPN or 4% WPN does not alter egg weights relative to control (control-1, control-2) eggs. However, there were significant treatment differences in hen FCR, total dozen eggs produced and feed intake (Table 4). Hens fed the control-2 diet produced significantly more dozen eggs than the control-1 hens, while hens fed the HOPN and WPN and control-2 produced similar numbers of total dozens of eggs over the 6-week feeding trial ($p < 0.05$). Hens fed the control-1 diet consumed significantly less feed relative to the other treatment groups over the 6-week feeding trial. Hens of the control-1 and WPN treatment groups had the lowest and most favorable feed conversion ratios, over the course of the 6-week feeding trial. Hence while hens fed the control-2 diet produced significantly more total dozen eggs compared to hens fed the control-1 diet, hens fed the control-1 diet consumed the least amount of feed compared (Table 4) to the other treatments, resulting in a significantly lower and improved FCR. Feed conversion ratios are important to the animal production industry as a measure of the kg of feed required to produce a dozen of eggs and serve as a tool for egg producers to maximize production profits.



**Table 3.** Body weights of hens fed a control or peanut-containing diet.

| | | | Treatments [1] | | | |
|---|---|---|---|---|---|---|
| | **Control-1** | **HOPN** | **WPN** | **Control-2** | **SEM** | ***p*-Value *** |
| | | | (kg) | | | |
| Week 0 | 1.64 | 1.64 | 1.67 | 1.66 | 0.02 | 0.07 |
| Week 2 | 1.57 | 1.58 | 1.60 | 1.57 | 0.02 | 0.42 |
| Week 4 | 1.58 | 1.66 | 1.65 | 1.62 | 0.04 | 0.32 |
| Week 6 | 1.61 | 1.64 | 1.66 | 1.64 | 0.03 | 0.46 |

[1] Dietary treatments: Control-1 = conventional diet containing defatted soybean meal and corn; HOPN = diet containing defatted soybean meal, corn, and 8% unblanched (skin intact) high oleic peanuts; WPN = diet containing defatted soybean meal, corn, and 4% whole in shell high-oleic peanuts; Control-2 = diet containing defatted soybean meal, corn, and soy protein isolate. 576 white Shaver laying hens (28 to 34 weeks of age) were assigned to one of 4 treatments with 4 replicates/treatment and provided feed and water ad libitum for 6-weeks. Body weights were recorded bi-weekly for each pen (18 hens per pen). Each value represents the mean pen weight ± the standard error. * *p*-value = statistically significant differences *p* < 0.05 by analysis of variance (ANOVA).

**Table 4.** Production performance per replicate of hens fed a control or peanut-containing diet [1].

| Treatments | Total Dozen Eggs Produced | Total Amt Feed Consumed (kg) | Total Amt Feed Consumed/Bird (kg/hen) | FCR (kg Total Feed Consumed/Total Dozen Eggs Produced) [2] | Egg Wt.(g) [3] |
|---|---|---|---|---|---|
| Control-1 | 120 [b] | 152 [b] | 4.23 [b] | 1.26 [b] | 57.1 |
| HOPN | 126 [a,b] | 169 [a] | 4.70 [a] | 1.34 [a] | 57.1 |
| WPN | 122 [a,b] | 163 [a] | 4.52 [a] | 1.26 [b] | 57.3 |
| Control-2 | 128 [a] | 161 [a] | 4.48 [a] | 1.33 [a] | 57.9 |
| SEM | 1.40 | 2.10 | 0.06 | 0.009 | 0.29 |
| *p*-value * | 0.01 | 0.0008 | 0.0008 | <0.0001 | 0.2061 |

[1] Dietary treatments: Control-1 = conventional diet containing defatted soybean meal and corn; HOPN = diet containing defatted soybean meal, corn, and 8% unblanched (skin intact) high oleic peanuts; WPN = diet containing defatted soybean meal, corn, and 4% whole in shell high-oleic peanuts; Control-2 = diet containing defatted soybean meal, corn, and soy protein isolate. There were 576 white Shaver laying hens (28 to 34 weeks of age) assigned to one of four treatments with four replicates/treatment and provided feed and water ad libitum for 6-weeks. Eggs were collected daily and enumerated weekly from each replicate pen (4 reps/treatment). Pen body weights were collected with ≈18 hens/pen. Feed intake was calculated weekly for each pen (≈18 hens/pen). [2] FCR = feed conversion ratio calculated as kg total feed consumed over the 6-week/total dozen eggs produced over 6-week feeding trial per replicate for each treatment (4 reps/treatment). Each value represents the replicate mean ± the standard error. * *p*-value = statistically significant differences *p* < 0.05 by analysis of variance (ANOVA). [3] Egg Wt = egg weights represent average egg weights per replicate for each treatment (4 reps/treatment). [a,b] Means within the same column lacking a common superscript differ significantly (*p* < 0.05).

Overall, there were no treatment differences in USDA grading or sizing of eggs between the controls (control-1, control-2) or peanut-containing treatment groups (WPN, HOPN) over the 6-week feeding trial. All eggs produced in this 6-week layer feeding trial had thick, firm egg whites and defect-free egg yolks. All eggshells were clean and without defects, with a minimal number of blood or meat spots (data not shown). Many of the eggs sub-sampled for quality, sizing and (120 eggs per treatment bi-weekly) USDA grading [26] were USDA grade A, large size eggs (Table 5), with greater than 96% Grade A eggs for each treatment. Eighty-six percent of the total 480 eggs sub-sampled (120 eggs sub-sampled bi-weekly) for the WPN treatment group were large size eggs, while approximately 9% were extra-large eggs. Yet, the other treatment groups had 89–90% of the 480 sub-sampled eggs categorized as large size eggs and 4–6% as extra-large eggs (Table 5). Most often when purchasing shell eggs from retail stores, egg sizes are determined by the total weight of the dozens of eggs produced into classifications of jumbo to small per the USDA grading shell egg standards [26].

**Table 5.** Size and USDA grading of shell eggs produced per replicate of hens fed a control or peanut-containing diet.

| | Treatments [1] | | | | | |
|---|---|---|---|---|---|---|
| | **Control-1** | **HOPN** | **WPN** | **Control-2** | | |
| | **(%) \*** | | | | **SEM** | ***p*-Value** |
| Grade A | 96.50 | 96.30 | 97.90 | 98.10 | 1.58 | 0.77 |
| Grade B | 1.88 | 3.13 | 0.00 | 0.00 | 1.58 | 0.45 |
| Cracks | 1.67 | 0.63 | 2.08 | 1.88 | 0.62 | 0.39 |
| Mechanical Loss | 0.00 | 0.00 | 0.21 | 0.00 | 0.10 | 0.43 |
| Extra-Large | 5.83 | 4.17 | 8.96 | 5.00 | 2.07 | 0.42 |
| Large | 89.00 | 90.00 | 86.04 | 90.83 | 1.24 | 0.08 |
| Medium | 4.38 | 5.42 | 3.96 | 3.96 | 1.52 | 0.89 |
| Small | 0.83 | 0.42 | 0.83 | 0.21 | 0.53 | 0.80 |

[1] Dietary treatments: Control-1 = conventional diet containing defatted soybean meal and corn; HOPN = diet containing defatted soybean meal, corn, and 8% unblanched (skin intact) high oleic peanuts; WPN = diet containing defatted soybean meal, corn, and 4% whole in shell high-oleic peanuts; Control-2 = diet containing defatted soybean meal, corn, and soy protein isolate. There were 576 white Shaver laying hens (28 to 36 weeks of age) assigned to one of four treatments (4 replicates/treatment). Bi-weekly (week 0, 2, 4, 6), a sub-sample of 120 eggs (30 eggs/rep) per treatment were assessed for USDA grading and sizing for a total of 480 eggs. \* Percentage of the 480 egg sub-sample per treatment.

In general, there were no significant differences in egg quality between the treatment groups, with exception of shell thickness at week 2 and yolk color at week 4 (Table 6). At the onset of the study (week 0), there were significant treatment differences in vitelline membrane elasticity and shell thickness (Table 6). Egg shell thickness was greatest in treatment WPN ($p < 0.0001$), and vitelline membrane elasticity was greatest in the controls ($p < 0.05$), relative to the other treatment groups at week 0. By week 2, there were no significant differences in vitelline membrane elasticity between the treatment groups. The vitelline membrane is a two-layer transparent protein matrix separating the egg yolk from the albumen and its integrity, strength, and elasticity are important quality characteristics required for separation of egg yolk and whites during egg breaking operations [27]. The mechanical properties of the vitelline membrane (strength, and elasticity) are commonly used to determine egg freshness [27]. Studies have shown a decline in vitelline membrane strength, and elasticity over time and is dependent upon storage conditions [28,29]. The egg albumen height is also commonly measured as an indicator of freshness and is measured as the height of the inner thick albumen of the egg when broken on a flat surface and declines with increasing storage time and freshness of the egg [30]. The HU (Haugh, 1937) is a measure of egg protein quality based upon the height of the albumen and the weight of the egg as an industry standard to measure egg quality and freshness [31], with the higher the HU the fresher the egg [20].

**Table 6.** Quality of eggs produced per replicate of hens fed a control or peanut-containing diet.

| | Treatments [1] | | | | | |
|---|---|---|---|---|---|---|
| | **Control-1** | **HOPN** | **WPN** | **Control-2** | **SEM** | ***p*-Value \*** |
| **Wk0** | | | | | | |
| Shell Sth. (g force) | 5298 | 5011 | 5346 | 5442 | 238 | 0.308 |
| SD (mm) | 0.29 [a] | 0.27 [a,b] | 0.25 [b] | 0.26 [a,b] | 0.01 | 0.001 |
| VMS (g) | 2.18 | 1.94 | 2.22 | 2.12 | 0.15 | 0.247 |
| VME (mm) | 1.70 | 1.41 | 1.78 | 1.68 | 0.18 | 0.205 |
| Shell Color (%) | 82.3 | 83.3 | 83.8 | 84.7 | 1.0 | 0.149 |
| Albumen Ht. (mm) | 8.15 | 8.49 | 8.22 | 8.45 | 0.29 | 0.593 |
| Haugh Unit (HU) | 90.9 | 92.5 | 91.0 | 92.3 | 1.6 | 0.614 |
| Yolk Color (1–15) | 1.88 | 2.17 | 2.21 | 2.21 | 0.21 | 0.335 |
| Shell Thick (mm) | 0.37 [c] | 0.38 [c] | 0.45 [a] | 0.44 [b] | 0.11 | <0.0001 |

**Table 6.** *Cont.*

| | | Control-1 | HOPN | WPN | Control-2 | SEM | *p*-Value * |
|---|---|---|---|---|---|---|---|
| **Wk2** | | | | | | | |
| | Shell Sth. (g force) | 5473 | 5350 | 5440 | 5432 | 247 | 0.97 |
| | SD (mm) | 0.235 | 0.232 | 0.241 | 0.241 | 0.01 | 0.34 |
| | VMS (g) | 2.45 | 2.40 | 2.45 | 2.22 | 0.18 | 0.18 |
| | VME (mm) | 1.96 | 1.94 | 2.0 | 1.75 | 0.21 | 0.64 |
| | Shell Color (%) | 83.3 | 81.6 | 83.9 | 83.2 | 0.86 | 0.06 |
| | Albumen Ht. (mm) | 8,85 | 8.47 | 8.44 | 8.73 | 0.32 | 0.46 |
| | Haugh Unit (HU) | 94.7 | 92.6 | 91.1 | 94.4 | 2.2 | 0.36 |
| | Yolk Color (1–15) | 2.83 | 2.50 | 2.83 | 2.88 | 0.17 | 0.10 |
| | Shell Thick (mm) | 0.39 [a] | 0.37 [b] | 0.39 [a] | 0.39 [a] | 0.01 | 0.03 |
| **Wk4** | | | | | | | |
| | Shell Sth. (g force) | 5466 | 5582 | 5428 | 5554 | 236 | 0.90 |
| | SD (mm) | 0.217 | 0.222 | 0.217 | 0.226 | 0.01 | 0.55 |
| | VMS (g) | 2.31 | 2.48 | 2.32 | 2.32 | 0.15 | 0.62 |
| | VME (mm) | 1.91 | 2.09 | 1.92 | 1.91 | 0.19 | 0.69 |
| | Shell Color (%) | 84.7 | 85.0 | 84.4 | 85.5 | 0.75 | 0.49 |
| | Albumen Ht. (mm) | 8.71 | 8.61 | 8.56 | 8.43 | 0.21 | 0.60 |
| | Haugh Unit (HU) | 93.5 | 93.0 | 92.6 | 92.0 | 1.10 | 0.58 |
| | Yolk Color (1–15) | 2.96 [a] | 2.67 [b] | 2.79 [b] | 3.08 [a] | 0.13 | 0.01 |
| | Shell Thick (mm) | 0.40 | 0.40 | 0.40 | 0.39 | 0.01 | 0.33 |
| **W6** | | | | | | | |
| | Shell Sth. (g force) | 5252 | 5164 | 5675 | 5282 | 251 | 0.18 |
| | SD (mm) | 0.214 | 0.226 | 0.226 | 0.217 | 0.01 | 0.56 |
| | VMS (g) | 2.29 | 2.40 | 2.17 | 2.20 | 0.14 | 0.39 |
| | VME (mm) | 1.89 | 2.02 | 1.75 | 1.77 | 0.18 | 0.42 |
| | Shell Color (%) | 83.7 | 84.1 | 83.4 | 84.3 | 0.72 | 0.56 |
| | Albumen Ht. (mm) | 8.34 | 8.26 | 8.50 | 8.33 | 0.24 | 0.79 |
| | Haugh Unit (HU) | 91.5 | 90.9 | 91.5 | 91.5 | 1.20 | 0.94 |
| | Yolk Color (1–15) | 2.92 | 2.50 | 2.54 | 2.79 | 0.17 | 0.046 |
| | Shell Thick (mm) | 0.39 | 0.38 | 0.40 | 0.39 | 0.01 | 0.22 |

There were 576 white shaver laying hens (28 to 34 weeks of age) assigned to one of four treatments with four replicates/treatment and provided feed and water ad libitum for 6-weeks. Per treatment, 24 eggs (6 eggs/replicate) were randomly selected and analyzed at each time point for quality assessment. [1] Dietary treatments: Control-1 = conventional diet containing defatted soybean meal and corn; HOPN = diet containing defatted soybean meal, corn, and 8% unblanched (skin intact) high oleic peanuts; WPN = diet containing defatted soybean meal, corn, and 4% whole in shell high-oleic peanuts; Control-2 = diet containing defatted soybean meal, corn, and soy protein isolate. Bi-weekly at 120 sub-sample of eggs were collected from each treatment group for quality assessment using Technical Services and Supplies QCD system, with calibration with the DSM Color Fan for yolk color. Yolk color = index 1–15 (lightest to darkest color intensity). Shell Sth = shell strength; SD = shell deformation; VMS = vitelline membrane strength; VME = vitelline membrane elasticity; Albumen Ht. = albumen height; Shell Thick = shell thickness. Each value represents the bi-weekly average ± the standard error with 120 eggs/treatment. * *p*-value = statistically significant differences *p* < 0.05 by analysis of variance (ANOVA). [a,b,c] Means within the same row lacking a common superscript differ significantly (*p* < 0.05).

Shell thickness was significantly less in eggs from the HOPN treatment group at week 2, relative to the other treatment groups (*p* < 0.05). By week 4 of the feeding trial, there were no significant treatment differences in any of the egg quality parameters measured except for yolk color. Egg yolk color was significantly greater in eggs produced from the control groups (control-1 and control-2) relative to the peanut-containing treatment groups (HOPN and WPN) at week 4 (*p* ≤ 0.01) of the feeding trial. At week 6 of the feeding trial, egg yolk color was greater in the control groups (control-1 and control-2) relative to

the peanut-containing treatment (HOPN and WPN) but was not statistically significant at $p < 0.05$.

In contrast, to our previous reports demonstrating that eggs produced from hens fed a 20% unblanched high-oleic peanut diet had significantly ($p < 0.0001$) enhanced egg yolk color in comparison to eggs produced from the controls when analyzed weekly in a 10-week and 8-week feeding trial, with no other effects seen on egg quality [6,10]. Moreover, in our previous feeding trials, there was approximately a 2-fold increase in yolk color of eggs produced from hens fed the unblanched high-oleic peanut diet compared to eggs of the controls ($p < 0.0001$) during the last four weeks of the study [6]. Feeding trials have demonstrated that egg yolk color has been shown to be greatly influenced by lipid profile (unsaturated vs. unsaturated and plant vs. animal dietary source of lipid) of the hen diet [32] and the type and concentration of dietary carotenoids with which are transferred along with their pigments to the yolks of the eggs produced [33,34]. Nevertheless, additional research is needed to determine precisely how dietary factors directly influence egg yolk color in layers.

At week 0 (prior to feeding experimental diets), eggs collected from the HOPN and WPN treatment groups had the highest levels of stearic fatty acid relative to control-1 and control-2 eggs (Table 7). Oleic fatty acid content was significantly reduced in eggs within the control-1 treatment group, relative to the other treatments at week 0. Elaidic acid content was significantly different between eggs of the control-1 and WPN treatment groups, while elaidic acid content was similar between eggs of treatments HOPN, WPN, and control-2 treatments at week 0. Linoleic fatty acid content was lowest in eggs of the control-2 treatment group, while linoleic fatty acid content was similar between the other treatment groups at week 0. Nervonic acid content was highest in eggs of the WPN and control-2 treatment groups relative to the control-1 and HOPN treatment groups at week 0. Lastly, omega 3 content was significantly different between eggs of the control-1 and HOPN experimental groups, with the lowest content in eggs of the HOPN treatment group at week 0. Omega 3 levels were similar between eggs of the control-1, WPN, and control-2 treatment groups at week 0.

At week 6 of the feeding trial, eggs produced from hens fed the HOPN diet had significantly reduced linoleic and stearic fatty acid content relative to the other treatment groups ($p < 0.0001$). This is also parallel to the analysis of the experimental diets, with the HOPN diet having the lowest levels of linoleic acid compared to the other treatment groups (Table 2). Linoleic acid is the major dietary source of omega 6 polyunsaturated fatty acid found in the Western diet and can be commonly found in vegetable oils, and oilseeds (sunflower, safflower, soybean). In past decades, nutritionists and health care professionals recommended a moderate daily intake of linoleic acid (10% of total energy intake) for the targeted prevention of cardiovascular disease [35]. However, currently, there is great debate within the scientific communities regarding the health benefits versus the potential adverse health impacts, such as increased production of pro-inflammatory compounds and increased risk for cancer, associated with increased dietary intake of linoleic acid [36]. Stearic acid, followed by palmitic acid, is one of the most commonly found saturated fatty acids within nature. Stearic acid is an 18-carbon chain found in meat, poultry, dairy products, lard, butter, beef tallow, cocoa butter, coconut oil, and palm oil [37]. In decades past, high intakes of dietary saturated fatty acids increased the risk of cardiovascular disease [37]. However, today new studies report that moderate intake of stearic acid does not increase the risk for cardiovascular disease and may reduce low density-lipoprotein cholesterol [38].

**Table 7.** The β-carotene, lipid and fatty acid analysis of eggs produced from hens fed a control or peanut-containing diet.

| | | Control | HOPN | WPN | Control-2 | SEM | *p*-Value * |
|---|---|---|---|---|---|---|---|
| | | | | | **Treatments [1]** | | |
| **Wk0** | | | | | | | |
| | Crude Fat | 5.23 | 4.33 | 5.28 | 6.36 | 0.70 | 0.061 |
| | Palmitic (16:0) | 22.9 | 23.2 | 23.3 | 22.2 | 0.36 | 0.28 |
| | Stearic (18:0) | 9.24 [b] | 9.58 [a,b] | 9.95 [a] | 9.33 [b] | 019 | 0.007 |
| | Oleic (18:1) | 28.4 [b] | 29.6 [a] | 29.8 [a] | 29.8 [a] | 0.48 | 0.030 |
| | Elaidic (C18:1trans) | 0.18 [a] | 0.09 [a,b] | 0.06 [b] | 0.10 [a,b] | 0.03 | 0.032 |
| | Linoleic (18:2) | 24.3 | 24.5 | 25.3 | 23.9 | 0.49 | 0.08 |
| | Nervonic (24:1, n − 9) | 1.03 [b] | 1.06 [b] | 1.12 [a] | 1.16 [a] | 0.03 | 0.002 |
| | Omega 3 (18:3) | 1.84 [a] | 1.55 [b] | 1.58 [a,b] | 1.71 [a,b] | 0.10 | 0.03 |
| | Cholesterol | 192 | 175 | 221 | 253 | 40 | 0.23 |
| | β-carotene | 2.02 | 1.99 | 2.19 | 2.03 | 0.22 | 0.69 |
| **Wk6** | | | | | | | |
| | Crude Fat | 6.09 | 7.08 | 7.31 | 5.80 | 0.94 | 0.32 |
| | Palmitic (16:0) | 21.4 | 21.2 | 21.3 | 21.8 | 0.50 | 0.64 |
| | Stearic (18:0) | 8.63 [a] | 7.03 [b] | 8.13 [a] | 8.36 [a] | 0.30 | <0.0001 |
| | Oleic (18:1) | 30.3 | 37.6 | 35.4 | 37.2 | 5.80 | 0.56 |
| | Elaidic (C18:1 trans) | 0.16 | 6.84 | 0.14 | 0.11 | 5.30 | 0.46 |
| | Linoleic (18:2) | 20.4 [a] | 11.8 [b] | 18.2 [a] | 17.4 [a] | 1.50 | <0.0001 |
| | Nervonic (24:1, n − 9) | 1.08 [a] | 0.73 [b] | 0.97 [a] | 0.88 [a,b] | 0.51 | <0.0001 |
| | Omega 3 (18:3) | 1.76 | 1.04 | 1.35 | 1.19 | 0.26 | 0.051 |
| | Cholesterol | 264 | 244 | 326 | 252 | 39 | 0.189 |
| | β-carotene | 2.41 [b] | 2.49 [b] | 3.82 [a] | 2.17 [b] | 0.57 | 0.043 |

There were 576 white Shaver laying hens (28 to 34 weeks of age) assigned to one of four treatments with four replicates/treatment and provided feed and water ad libitum for 6-weeks. Sixteen eggs/treatments were chemically analyzed at each time point of collection. [1] Dietary treatments: Control-1 = conventional diet containing defatted soybean meal and corn; HOPN = diet containing defatted soybean meal, corn, and 8% unblanched (skin intact) high oleic peanuts; WPN = diet containing defatted soybean meal, corn, and 4% whole in shell high-oleic peanuts; Control-2 = diet containing defatted soybean meal, corn, and soy protein isolate. At week 0 (wk0) and week 6 (wk6) of the feeding trial, 4 eggs were randomly selected per replicate (16/treatment) and chemically analyzed for lipid content (cholesterol, crude fat), fatty acid, and β-carotene content by an AOAC-certified lab (ATC Scientific, Little Rock, AR, USA), using AOAC approved standard methods. Each value represents the mean ± the standard error with 16 eggs/treatment. * *p*-value = statistically significant differences $p < 0.05$ by analysis of variance (ANOVA). [a,b] Means within the same row lacking a common superscript differ significantly ($p < 0.05$).

Nervonic fatty acid content was similar between eggs produced from hens fed the control-1, WPN, and control-2 dietary treatments (Table 7), while the nervonic fatty acid content was significantly reduced in eggs produced from hens fed the HOPN diet relative to eggs of the control-1 group ($p < 0.0001$). However, nervonic acid levels were very low (0.2 g/kg dry matter) and similar between all experimental diets (data not shown). Nervonic acid is a very long (24 carbon chain) monounsaturated omega 9 fatty acid, with its name derived from its discovery within mammalian nerve tissue. In vivo, nervonic acid combines with sphingosines to form sphingolipids which are important components of brain white matter and thus important to neural health [39]. Nervonic acid is found in some seed oil of some wild plants but is also biosynthesized by elongation from oleic acid within the endoplasmic reticulum membrane [39]. While there were no statistically significant treatment differences in total omega 3 content between the eggs produced at week 6 of the study ($p = 0.051$), eggs produced from hens fed the control-1 diet had the highest and eggs produced from hens fed the HOPN diets had the lowest levels of omega 3 content. Interestingly, chemical analysis of the experimental diets (Table 2) demonstrated that total omega 3 content was highest in both control experimental diets and lowest in HOPN (control-1, 63.2 g/kg; control-2, 65.3 g/kg; WPN, 46.1 g/kg; HOPN, 15.1 g/kg dry matter)

β-carotene levels were significantly enhanced in eggs produced from hens fed the WPN treatment group, relative to the other treatments at week 6 of the feeding trial ($p < 0.05$). In contrast, β-carotene levels were below the assay threshold level of detection (<5 ppm) in the chemical analysis of all experimental diets (data not shown). Studies have reported low levels of β-carotene in unrefined extracted soybean oil [40]. Nevertheless, these low levels of β-carotene are lost with extraction and refinement of commercially manufactured soybean oil, utilized in preparation of control-1, WPN and control-2 experimental diets. Extracted peanut oil has been reported to contain small quantities of β-carotene [41]. Pattee and Purcell, (1967) demonstrated that peanut oil extracted from young peanuts contained 60 μg of β-carotene and 138 μg of lutein per liter, while peanut oil extracted from more mature peanuts had lower concentrations [42]. Peanuts have an indeterminate growth pattern, and thus at harvest peanut pods are collected from a range of differing maturity levels on the peanut plant. Harvest containing a higher percentage of young pods may have elevated levels of β-carotene and lutein in the seed of peanuts. Hence, the whole-in-shell peanuts (WPN) utilized in this study may potentially have had a higher percentage of young peanuts containing elevated levels of β-carotene which enriched the eggs produced in this treatment group, while the unblanched high-oleic peanuts utilized in this study may have had a lower percentage of young peanuts containing β-carotene. In contrast, our previous feeding trials demonstrated that eggs produced from hens fed a HOPN diet had enhanced β-carotene content and yolk color relative to the control eggs [6,10]. Also, in our previous studies, eggs produced from hens fed a 20% unblanched high-oleic peanut diet had approximately a 1.5-fold increase in β-carotene content relative to the control eggs at week 5 and week 10 of the feeding trial ($p < 0.0001$) [6], which may have been due to the use of younger β-carotene-containing peanut pods in the experimental diet.

## 4. Conclusions

Most importantly, this study reports similar body weights, feed intake, and egg weights between the peanut-containing diets (4% WPN and 8% HOPN diets) to both control diets implying the effective utilization of whole-in-shell high-oleic peanuts and/or unblanched high-oleic peanuts as alternative layer feed ingredients at these inclusion rates. This study supports the use of peanuts as a value-added feed ingredient to support poultry and egg production within North Carolina and the US Southeast and agricultural sustainability. Moreover, we demonstrated that hens of the non-conventional control treatment group produced more total dozen eggs than the conventional control group. However, hens of the conventional control and WPN treatment groups had the best feed conversion ratio compared to the other treatment groups, suggesting that the use of a conventional control diet containing defatted soybean meal + corn is a more suitable dietary control for future layer feeding trials. While this study has positive implications for the use of peanuts (unblanched or whole in-shell) as an alternative layer feed ingredient, this feeding trial must be replicated to more closely parallel commercial egg production methods.

**Author Contributions:** Authors K.L.H., R.W. and T.V. were actively involved in the care and husbandry of all research animals, collection, and analysis of eggs; and co-authors K.L.H., O.T.T., R.M. and K.E.A. were active participants in the data analysis, data interpretation, and preparation of the manuscript. All authors have read and agreed to the published version of the manuscript.

**Funding:** This work was funded by the Food Science & Market Quality and Handling Research Unit, Agricultural Research Service (Project 6070-43440-013-000D), and the North Carolina Peanut Growers Association (Proposal 2021-0561, Project ID 572099-87361).

**Institutional Review Board Statement:** The procedures used in these studies were approved by the North Carolina State University Institutional Animal Care and Use Committee (IACUC #19-761-A, approved 11/27/2019, expires 11/27/2022).

**Informed Consent Statement:** Not applicable. No human subjects were used in this study and therefore consent was not required.

**Data Availability Statement:** The data presented in this study are available on request from the corresponding author.

**Acknowledgments:** The authors would gratefully like to acknowledge the following: North Carolina Peanut Growers' Association and staff of the Prestage Department of Poultry Science-North Carolina State University, staff of North Carolina State University Feed Mill, Birdsong Peanuts and Jimbos Jumbos for the donation of peanuts for these feeding trials, and the Food Science & Market Quality Handling Research Unit-ARS for their contributions to this study. This work was supported by funds from the North Carolina Peanut Growers Association.

**Conflicts of Interest:** There is absolutely no conflict of interest regarding this manuscript.

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
