# Peer review of "The Effects of Feeding a Whole-in-Shell Peanut-Containing Diet on Layer Performance and the Quality and Chemistry of Eggs Produced"

_agriculture, doi:10.3390/agriculture11111176_

Round 1

Reviewer 1 Report

My comments has been attached

Author Response

Reviewer 1-Agriculture Journal-Whole In Shell Peanut Layer Manuscript

  1. Is 8% out of 99% all that important? What does the 99% stand for? Please recast the sentence.

Author Response: Sentence re-phrased to clearly state that NC is one of the top 6 peanut producing states in the US, which produces nearly all the US annual peanut crop citing a new reference.

  1. This doesn't describe the experimental design, please.

Author Response: Per the reviewer’s comments, the 2.1 sub-section was retitled to “2.1 Experimental Design, Animal Husbandry and Dietary Treatments”. Additionally, the experimental design of how hens were allocated to the different groups is stated clearly in the first paragraph of this section.

  1. ME or GE? Please check you Tables on diet composition.

Author Response: In sub-section 2.1, it has been edited to clearly state the diets were formulated to be isocaloric (2,928 kcal/kg metabolizable energy). Edits were made to clearly state ME in table 1 legend and in the body of the text in section 2.1.

  1. Please read line 118 -119. GE is not important. You have already provided the ME values of the diets.

Author Response: In table 2, the crude protein and the gross energy for each of the finished feed samples was removed in the edited manuscript per the reviewers’ comments. In sub-section 2.1 and table 1, it has been edited to clearly state the diets were formulated to be isocaloric (2,928 kcal/kg metabolizable energy). All values in table 2 were double-checked for accuracy and edited, in addition, the body of the text of the results for table 2 were also edited accordingly.

  1. What is the measure of "greater" here? Please recast the sentence.

Author Response: For clarity, this statement was added to the body of the text results section of the revised manuscript.

“Egg yolk color was significantly greater in eggs produced from the control groups (control-1 and control-2) relative to the peanut-containing treatment groups (HOPN and WPN) at week 4 (P≤0.01)  of the feeding trial. Also, at week 6 of the feeding trial, egg yolk color was greater in the control groups (control-1 and control-2) relative to the peanut-containing treatment (HOPN and WPN) but was not statistically significant at P<0.05.”

In addition, the assigned superscripts indicating the statistical differences were removed at week 6 of table 6.

  1. This Table doesn't mean much as presented. You need to analyse the data.

  1. All these values are already in Table 5. Please avoid duplication as much as possible.

Author Response: In table 5 of the manuscript, we provided the actual number of eggs in egg size and USDA egg category from the 120 egg sub-sample collected bi-weekly from each treatment group. As a consequence, we only mention these values as percentages, in the body of the text of the results to discuss these results in 3 separate sentences, to avoid duplication.

  1. Please note that you did not provide any details of the peanut shell used for the study

Author Response: In lines 120 to 132 of the revised manuscript, we discuss the details of the whole in-shell peanut in the preparation of the experimental diet. Please see this section pasted below with direct reference to the whole in shell peanut diet in red font for better clarity.

Line 120 to 132 of revised manuscript: “The whole-in-shell peanut diet (WPN) was prepared using 8% whole-in-shell high-oleic peanuts + solvent extracted defatted soybean meal + yellow corn, and the unblanched high oleic peanut diet (HOPN) was prepared using 4% unblanched (skin intact) high-oleic peanuts + solvent extracted defatted soybean meal + yellow corn.  Aflatoxin-free peanuts were used in all peanut-containing experimental diets. Whole-in-shell high-oleic peanuts were chemically analyzed by ATC Scientific (Little Rock, AR, USA) for nutritional content.  The nutritional content for whole-in-shell high-oleic peanuts was determined to be: 34.8% crude fat, 22.0% crude protein, 0.16% calcium, 0.30% phosphorous, 34.2% carbohydrates, <5ppm β-carotene, gross energy 6500 kcal/kg. Whole-in-shell high-oleic peanuts and unblanched high-oleic peanuts were crushed using a Roller Mill to form crumbles, prior to inclusion in the finished diets. Each of the experimental diets were supplemented with vitamin, mineral, and selenium premixes manufactured at the NC State University Feed Mill (Raleigh, NC, USA) to meet and/or exceed poultry requirements for vitamins, minerals, and selenium.”

  1. I am unsure if this can be easily replicated due to poor description of the experimental procedure.

Author Response: We have previously conducted and successfully published 7 poultry feeding trials (4 laying hens and 3 broiler chicken) utilizing peanuts as an alternative feed ingredient, using very similar experimental design, experimental diet preparation, sampling procedures and methods. The citations for these publications are listed below in blue font.

Author Response: Per the reviewers’ comments the manuscript was edited for clarity. The body of the text of the materials and methods was edited to clearly state the experimental design, animal husbandry and methods in sub-section 2.1. Please see these edited sections in red font of the revised manuscript and pasted below. Subsequently, lines 117 to 138, in the body of the text detail the preparation of the experimental diets, line 140 to161 in the body of the text provides detailed methods conducted to determine egg quality and grading, lines 162 to 177 provides detailed methods for conducting egg chemical analysis and lines 179 to 191 provides details of the statistical analysis.

2.1 Experimental Design, Animal Husbandry and Dietary Treatments

This study was conducted at the North Carolina Department of Agriculture and Consumer Services Piedmont Research Station (Salisbury, NC, USA). Four experimental diets were formulated in Concept 5 (level 2, version 10.0) to be isocaloric (2,928 kcal/kg metabolizable energy) and isonitrogenous (19.5% crude protein) with an estimated particle size between 800 and 1000 µm (Table 1). Five hundred seventy-six Shaver laying hens (28 to 34 weeks of age) were randomly assigned to one of the four dietary treatments (144 hens per treatment), with four replicates of thirty-six birds per treatment. Hens were housed with 18 birds per cage allowing a space of 69 in2 per hen. Cages consisted of two rows of a Conventional Tri-Deck Stacked Layer Cage System with 26 x 48 in2 per cage. The study was conducted in a standard height, windowless enclosed ventilated house.

Throughout the feeding trial, birds were provided 14 L:10 D and feed and water ad libitum for 6-weeks. Pen body and feed weights were recorded bi-weekly. Shell eggs were collected and enumerated daily from each pen and replicate and totaled each week. Total number of eggs produced per replicate for each treatment was calculated for the total 6 week feeding trial. The average feed conversion ratio (FCR) was calculated as total feed consumed over the 6-week feeding trial (kg)/total of dozens eggs produced for each treatment group over the 6-week feeding trial.  

Toomer, O.T., Vu, T.C., Wysocky, R., Moraes, V., Malheiros, R., Anderson, K.E. 2021. The effect of feeding egg-producing hens a peanut skin-containing diet on performance, shell egg quality and lipid chemistry. Agriculture. 11(9): 10.3390/agriculture11090894

Toomer, O.T., Vu, T.C., Sanders, E., Redhead, A.K., Malheiros, R. and Anderson. K.E. Feeding Laying Hens a Diet Containing High-Oleic Peanuts or Oleic Acid Enriches Yolk Color and Beta-Carotene While Reducing the Saturated Fatty Acid Content in Eggs. Agriculture. 2021; 11(8): 10.3390/agriculture11080771

Redhead, A.K., Sanders, E., Vu, T.C., Malheiros, R.D., Anderson, K.E. and Toomer, O.T. The effects of high-oleic peanuts as an alternate feed ingredient on performance, ileal digestibility, apparent metabolizable energy, and histology of the small intestine in laying hens. Transl. Anim. Sci. 2021; 5(1):txab015. doi: 10.1093/tas/txab015.

Toomer, O.T., Livingston, M., Wall, B., Sanders, E., Vu, T., Malheiros, R.D., Livingston, K.A., Carvalho, L.V., Ferket, P.R. and Dean, L.L. Feeding high-oleic peanuts to meat-type broiler chickens enhances the fatty acid profile of the meat produce. Poult. Sci. 2020; 99(4):2236-2245. doi: 10.1016/j.psj.2019.11.015.

Toomer, O.T., Sanders, E., Vu, T.C., Malheiros, R.D., Redhead, A.K., Livingston, M.L., Livingston, K.A., Carvalho, L.V. and Ferket, P.R. The effects of high-oleic peanuts as an alternative feed ingredient on broiler performance, ileal digestibility, apparent metabolizable energy, and histology of the intestine. Transl. Anim. Sci. 2020; 4(3):txaa137. doi: 10.1093/tas/txaa137.

Toomer, O.T., Hulse-Kemp, A.M., Dean, L.L., Boykin, D.L., Malheiros, R. and Anderson. K.E. Feeding high-oleic peanuts to layer hens enhances egg yolk color and oleic fatty acid content in shell eggs. Poult. Sci. 2019; 98(4):1732-1748.

Toomer, O.T., Livingston, M., Wall, B., Sanders, E., Sipple, L., Vu, T., Malheiros, R.D., Livingston, K.A., Carvalho, L.V., Ferket, P.R. and Drake MA. Meat quality and sensory attributes of meat produced from broiler chickens fed a high oleic peanut diet. Poult. Sci. 2019; 98(10):5188-5197. doi: 10.3382/ps/pez258.

Reviewer 2 Report

Dear Authors,

Please see my comments:

  • Line 107. I would recommend you to change the cage size unit to a centimeter or meter. I understand the target of this study is mostly US consumers, however, readers from other countries might be interested as well. Please check the whole manuscript. 
  • There are minor issues with spacing, formatting or typos. Please check the whole manuscript.
  • Table 1. Fix it.
  • Section 3.2. The section requires adding a few more lines and discussing the results as well as a comparison between this study and others.  
  • Section 3.4. line 490. citation needs to be fixed. Also, this section requires more citations, and some citations are from 1967, add most recent works as well.  

Author Response

Whole In Shell Layer Manuscript-Reviewer 2 Comments

Comments and Suggestions for Authors

Dear Authors,

Please see my comments:

  • Line 107. I would recommend you to change the cage size unit to a centimeter or meter. I understand the target of this study is mostly US consumers, however, readers from other countries might be interested as well. Please check the whole manuscript. 

Author Response: This sentence was edited to read as the following: Hens were housed with 18 birds per cage allowing a space of 175.26 cm2 per hen. Cages consisted of two rows of a Conventional Tri-Deck Stacked Layer Cage System with 66.04 x 121.92 cm2 (26 x 48 in2) per cage.

  • There are minor issues with spacing, formatting or typos. Please check the whole manuscript.

Author Response: Per the reviewer’s comments, we ensured that only one space was provided after each sentence throughout the manuscript. Subsections in the Materials and Methods section were numbered appropriately (2.1 Experimental Design, Animal Husbandry and Dietary Treatments, 2.2 Egg Quality and Grading, 2.3 β-Carotene, lipid and fatty acid analysis, 2.4 Statistical analysis). Moreover, spell check was performed on the entire manuscript and no spelling errors were found. All punctuation errors in “Word Editor” were also corrected.

  • Table 1. Fix it.

Author Response: Per the reviewers’ comments, top and bottom borders were added to the table 1 for improved clarity and distinguishment of the content between the columns and rows.

  • Section 3.2. The section requires adding a few more lines and discussing the results as well as a comparison between this study and others.  

Author Response: Additional lines were added in section 3 to discuss our previous findings to correlate with the findings of this study. Please see the additional lines added in this section “in tracking”.

  • Section 3.4. line 490. citation needs to be fixed. Also, this section requires more citations, and some citations are from 1967, add most recent works as well.  

Author Response: :Unfortunately, there were 2 publications (Al Juhaimi et al. 2018 and Pattee and Purcell, 1967) that I have been able to find as reference to the beta-carotene content in peanut oil, and only 1 publication regarding the carotenoid content in peanut oil as related to peanut maturity (Pattee and Purcell 1967). As a consequence, I have utilized these references, which are the only ones that I could find in this section of the manuscript. I have pasted a web capture of one of my searches as reference below. These references are utilized as a means to potentially explain the elevation of beta-carotene in the eggs produced from the WPN treatment group. Reference Al Juhaimi et al. 2018 was added to the reference list of the manuscript.

Reviewer 3 Report

General comments

The paper deals with the  effect of diets containing whole-in shell peanut on layer performance and quality and physicichemical composition of eggs produced.  This is a very interesting paper because the use of food industry by-products for the production of feed ingredients in circular-economy schemes is a very hot topic. The paper can provide some new information about the use of these by-products in layers´feeding since there is few information about the use of the whole-in-shell product. The work is within the scope of the journal and presented some novelty. The manuscript in general is sound, experimental desing is correct but authors should improve thoroughly the results and discussion section.

Specific comments

Introduction

L82: include a number for the reference of Aka et al (2020)

The introduction section succed in explain the background information of the topic but the objetives of the work should be more clearly explained. As it was explained in the text the first objetive was to determine the effect of whole-in-shell peanuts on performance and production as well as on the lipid and fatty acid content of produced eggs. The second objetive was to compare the two controls. This second objetive is not very clear since the whole manuscript deals with the use of peanuts’ by-products. Moreover, it is not clear why the HOPN treatment was include, what was the objetive of including it?. The discussion of the manscript should take into account the objetives stated in the introduction section.

Material and Methods

L103-105: this sentence is repeated in L98-100, check the number of the aproval because is different in L98-100 compared to L103-105

L124: 4%  instead of 8%

L126: 8% instead of 4%

L129 and L133-135: delete references to dehydrated sweet potato by-products

L129: please indicate the methods used to do the analyses with the corresponding reference and not only the laboratory

L135: The diets were in form of pellets? Indicate

L145, methods numbers should be added and a reference for AOAC. Include also the method for other parameters shown in Table 2 as crude protein, phosphorous, calcium, gross energy

L160: please check the reference format for (Haugh, 1937)

L173: Why 144 eggs? If there were 16 egg/treatment x 4 treatments x 2 time measurements

L191: why some of the measurements were analysed by week and not taking into account in the statistical model the effect of the time? Or using repeated measurements?; Moreover, indicate the statistical model used, what were the fixed effects , the random if exist, …

L 233: Table 1: use g/kg DM as unit for ingredients

L247: Table 2: use g/kg DM as unit for composition; in the footnote delete from “Four dietary treatments……AOAC-approved methods”. In the footnote delete “2” and “*” and include the units in the table (i.e. crude fat (g/ kg DM) or Oleic (g Fatty acid/100 g total lipid content). I suggest also to include the chemical formula for the fatty acids instead of the common name (i.e. C18:0 for stearic, etc)

Results and Discussion

Results and discussion section should be thoroughly improved. Authors did not succed in discussing scientifically the observed findings in a biologically integrated fashion, both within the study as well as relative to results of other scientists. Little discussion of the results appeared in this section, mainly appered results description (i.e. L333-349; L355-365,…)

L217: 2”inflamatory” or “anti-inflamatory”?

Table 3 and 4: please include SEM in a different colum

L384 and L388: check the P-values with the values of the table

L391: Discuss changes observed in yolk color, why did occur?, other authors observed this changes?,….

L393-401: delete or reduce, since taking into account your results it is not a very relevant information

Table5: There no statistical analyses on this data? No SEM or P-value?

Table 6: include SEM in a different colum. Revise superscripts for shell thick in week 2, same means have different superscripts (Control 1 and WPN and Control-2)

Table 7: include units for each item; include SEM in a different colum; Preferably use the chemical formula for the fatty acids instead of the common name (i.e. C18:0 for stearic, etc)

When discussing the effect of the diets on lipid profile of eggs, authors should mention the lipid profile of the different diets. As it can be seen in table 1, all diets except for HOPN had soybean oil as fat source, being this oil mainly composed of linoleic acid and with a relatively high proportion of linolenic acid. These differences might have an effect on the observed results and authors have to mention and discuss about it

References

As along the manuscript some references had not the correct format, check if all the references cited in the manuscript appeared in the Reference list

Author Response

General comments

The paper deals with the  effect of diets containing whole-in shell peanut on layer performance and quality and physicichemical composition of eggs produced.  This is a very interesting paper because the use of food industry by-products for the production of feed ingredients in circular-economy schemes is a very hot topic. The paper can provide some new information about the use of these by-products in layers´feeding since there is few information about the use of the whole-in-shell product. The work is within the scope of the journal and presented some novelty. The manuscript in general is sound, experimental design is correct but authors should improve thoroughly the results and discussion section.

 Author response: Thank you.

Specific comments

Introduction

L82: include a number for the reference of Aka et al (2020)

Author response: Per the reviewer’s comments the reference number was added.

The introduction section succeed in explain the background information of the topic but the objetives of the work should be more clearly explained. As it was explained in the text the first objetive was to determine the effect of whole-in-shell peanuts on performance and production as well as on the lipid and fatty acid content of produced eggs. The second objetive was to compare the two controls. This second objetive is not very clear since the whole manuscript deals with the use of peanuts’ by-products. Moreover, it is not clear why the HOPN treatment was include, what was the objetive of including it?. The discussion of the manscript should take into account the objetives stated in the introduction section.

Author response: The 3 objectives in the introduction were more clearly defined in the introduction in the revised manuscript. “we aimed to determine the effect of whole-in-shell peanuts and unblanched peanuts on the performance and production of laying hens. Secondly, we aim to determine, as well as the effects feeding a whole-in-shell peanut diet or unblanched peanut diet has on the quality, lipid and fatty acid content of the eggs produced.” “As a consequence, we aim to compare the effects of layer production performance, egg quality and chemistry of hens fed the peanut-containing diets (whole in-shell peanut and unblanched peanut) to a conventional industry standard control diet prepared with defatted soybean meal and yellow corn and a control diet prepared with soy protein isolate, defatted soybean meal and yellow corn to validate these previous findings.” 

Author response: Also, additional details were provided in the introduction to justify the research objectives in this study. Please see these edits in the “tracking” feature in word.

Material and Methods

L103-105: this sentence is repeated in L98-100, check the number of the aproval because is different in L98-100 compared to L103-105

Author response: This was an inadvertent error and was corrected to state the following: “All animal research procedures used in these feeding trials were approved by the North Carolina State University Institutional Animal Care and Use Committee (IACUC #19-761-A).”

L124: 4%  instead of 8%

Author response: This inadvertent error was corrected in the revised manuscript, and can be found in the “tracking” feature of word. The edited sentence states the following, “The whole-in-shell peanut diet (WPN) was prepared using 4% whole-in-shell high-oleic peanuts + solvent extracted defatted soybean meal + yellow corn, and the unblanched high oleic peanut diet (HOPN) was prepared using 8% unblanched (skin intact) high-oleic peanuts + solvent extracted defatted soybean meal + yellow corn.”

L126: 8% instead of 4%

Author response: Author response: This inadvertent error was corrected in the revised manuscript, and can be found in the “tracking” feature of word. The edited sentence states the following, “The whole-in-shell peanut diet (WPN) was prepared using 4% whole-in-shell high-oleic peanuts + solvent extracted defatted soybean meal + yellow corn, and the unblanched high oleic peanut diet (HOPN) was prepared using 8% unblanched (skin intact) high-oleic peanuts + solvent extracted defatted soybean meal + yellow corn.”

L129 and L133-135: delete references to dehydrated sweet potato by-products

Author response: Reference to “sweet potato by-products” was removed from the revised manuscript.

L129: please indicate the methods used to do the analyses with the corresponding reference and not only the laboratory

Author response: Per the reviewer’s comments this line was edited to the following sentence, “The nutritional content for whole-in-shell high-oleic peanuts was determined to be: 34.8% crude fat, 22.0% crude protein, 0.16% calcium, 0.30% phosphorous, 34.2% carbohydrates, <5ppm β-carotene, gross energy 6500 kcal/kg using standard AOAC-approved methods for nuts and seeds (crude fat determination using Gravimetric methods for nuts-AOAC 948.22, protein determination using Kjeldahl method for nuts-AOAC 950.48, mineral determination by elemental analysis of mineral by atomic absorption spectroscopy, carbohydrates were determined using standard colorimetric assay determination and spectroscopy, enzymatic-gravimetric methods were used for carbohydrate determination-AOAC 991.43, standard bomb calorimetry methods were used to determine gross energy, and β-carotene was determined using standard high-performance liquid chromatography and spectrophotometry methods).” in the revised manuscript. Please see these changes in the “track changes” feature in word.

L135: The diets were in form of pellets? Indicate

Author response: All 4 experimental diets were fed as “mash” diets and not pelleted. This was included in the revised manuscript in the following line, “All four experimental diets were fed as mash diets and were analyzed by the North Carolina Department of Agriculture and Consumer Services and the Food and Drug Protection Division Laboratory (Raleigh, NC, USA) for aflatoxin and microbiological contaminants.”

L145, methods numbers should be added and a reference for AOAC. Include also the method for other parameters shown in Table 2 as crude protein, phosphorous, calcium, gross energy

Author response: Per the reviewer’s comments the following line was edited as the following: “All experimental diets  were analyzed for crude fat, total cholesterol, fatty acid profile, and β-carotene by an AOAC-certified lab, ATC Scientific (Little Rock, AR, USA), using AOAC approved standard methods described by Toomer et al. (2019 ).”

L160: please check the reference format for (Haugh, 1937)

Author response: Per the reviewer’s comments, this line was corrected in the revised to the following line: “Haugh Unit and albumen height were analyzed using the TSS QCD System (Technical Services and Supplies, Dunnington, York, UK). HU is calculated using the following calculation=100Log (h-1.7w + 7.6), with h=egg albumen height and w=weight of egg, with values ranging from 0 to 130 and HU scores below 60 for un-fresh eggs [17].”

L173: Why 144 eggs? If there were 16 egg/treatment x 4 treatments x 2 time measurements

Author response: This error was corrected to read, that a total of 128 eggs (64 eggs at each time point, with 16 eggs/treatment with 4 treatments. Please see these changes in “track changes” in word in the revised document.

L191: why some of the measurements were analysed by week and not taking into account in the statistical model the effect of the time? Or using repeated measurements?; Moreover, indicate the statistical model used, what were the fixed effects , the random if exist, …

Author response: The statistical design of this study did not account for experimental or treatment differences over time. In our previous published high-oleic peanut layer feeding trials demonstrated highly significant differences in variables measured over time, which can be attributed to dietary adaptation to the experimental diets over the weekly course of the feeding experiment. However, our interest in this study was/is treatment differences in the variables, and not statistical differences bi-weekly. We reported the data in Table 4 as a value representative of the entire feeding trial (6 weeks), which is preferred and most useful to the egg production industry. All other variables measured (body weights, egg quality, egg size and grading) were measured bi-weekly, which is more customary in layer feeding trials. Egg chemistry was determined at week 0 and week 6, due to cost of egg analysis, which was cost-prohibitive to analyze additional time points.

Author response: For clarity of the results, we added the statement in the revised manuscript, “There were no significant differences in the variables measured between treatment replicates.” Please see this line in the revised manuscript in “tracking”. Please see the pasted images of tables from previous publication.

L 233: Table 1: use g/kg DM as unit for ingredients

Author response: Per the reviewer’s comments Table 1 has been edited to use g/kg DM units for ingredients. Please see the revised manuscript.

L247: Table 2: use g/kg DM as unit for composition; in the footnote delete from “Four dietary treatments……AOAC-approved methods”. In the footnote delete “2” and “*” and include the units in the table (i.e. crude fat (g/ kg DM) or Oleic (g Fatty acid/100 g total lipid content). I suggest also to include the chemical formula for the fatty acids instead of the common name (i.e. C18:0 for stearic, etc)

Author response: Per the reviewer’s comment Table 1 and Table 2 ingredients list unit were edited to g/kg DM. The line containing “Four dietary treatments…….AOAC-approved methods” in the footnotes for Table 2 was deleted. Also, “information in the footnote of Table 2 regarding the units were deleted. The chemical formula with the number of carbons and double bonds was included in the revised manuscript.

Results and Discussion

Results and discussion section should be thoroughly improved. Authors did not succed in discussing scientifically the observed findings in a biologically integrated fashion, both within the study as well as relative to results of other scientists. Little discussion of the results appeared in this section, mainly appered results description (i.e. L333-349; L355-365,…)

Author response:

L217: 2”inflamatory” or “anti-inflamatory”?

Author response:This error is corrected in the revised manuscript in “track changes” to read “anti-inflammatory”. Please see the revised manuscript.

Table 3 and 4: please include SEM in a different colum, Table 6: include SEM in a different colum, Table 7: include SEM in a different colum

Author response: Per the reviewer’s comments tables 3, 4, 5, 6 and 7 the SEM was placed in a separate column in the revised manuscript.

L384 and L388: check the P-values with the values of the table

Author response: The inadvertent error with the superscripts at week 2 shell thickness was corrected in Table 6. Also, we ensured that the p-values present in the body of the text correspond with the values in Table 6. This paragraph was edited in “tracking” to correct the p-values in the body of the text.

L391: Discuss changes observed in yolk color, why did occur?, other authors observed this changes?,….

Author response: For clarity this line has been edited to indicate why these changes are seen in yolk color. Please see the following edited sentence: “Feeding trials have demonstrated that egg yolk color is greatly influenced by lipid profile of the hen diet [29] and the type and concentration of dietary carotenoids with which are transferred along with their pigments to the yolks of the eggs produced[30, 31]” in the revised manuscript.

L393-401: delete or reduce, since taking into account your results it is not a very relevant information

Author response: These lines in the “results” section of the body of the text were deleted. Alternatively, more discussion was provided to parallel our previous high oleic peanut layer feeding trial to the current study (Toomer et al. 2019). Please see the revised manuscript in “tracking”.

Table5: There no statistical analyses on this data? No SEM or P-value?

Author response: This data table was edited to represent the percentages of each egg USDA and sizing category with the statistical analysis in the revised manuscript. Please see the revised Table 5 manuscript. The footnote of the table 5 was also edit. Additionally, the body of the text was edited accordingly in the revised manuscript.

Revise superscripts for shell thick in week 2, same means have different superscripts (Control 1 and WPN and Control-2)

Author response: The superscripts were corrected in that the control-1, control-2, and WPN assignment as “a” superscript, and HOPN assigned the “b” superscript.

Table 7: include units for each item; include SEM in a different colum; Preferably use the chemical formula for the fatty acids instead of the common name (i.e. C18:0 for stearic, etc)

Author response: Per the reviewer’s comments the SEM was put in a separate column in table 7 and the formula for the fatty acids, with number of carbons and double bonds were also added in table 7 of the revised manuscript.

When discussing the effect of the diets on lipid profile of eggs, authors should mention the lipid profile of the different diets. As it can be seen in table 1, all diets except for HOPN had soybean oil as fat source, being this oil mainly composed of linoleic acid and with a relatively high proportion of linolenic acid. These differences might have an effect on the observed results and authors have to mention and discuss about it

Author response: Per the reviewer’s comments, discussion of the fatty acid content of the experimental diets were discussed with the fatty acid content of the eggs as “in parallel” or “in contrast”. Please see the revised manuscript.

References

As along the manuscript some references had not the correct format, check if all the references cited in the manuscript appeared in the Reference list

Author response: All citations in the body of the text were cross reference against the reference list to ensure for accuracy. Please see the revised manuscript.

Round 2

Reviewer 1 Report

Well revised. 

Authors need to read through again carefully and improve the language, especially in the discussion. example below:

The formulation of the diets using the analyzed values for the WPN and HOPN in- 1992 gredients were calculated to be isocaloric and isonitrogenous (Table 1). 

Author Response

Reviewer comment: Well revised. 

Author response: Thank you.

Reviewer comment: Authors need to read through again carefully and improve the language, especially in the discussion. example below:

The formulation of the diets using the analyzed values for the WPN and HOPN in- 1992 gredients were calculated to be isocaloric and isonitrogenous (Table 1). 

Author response: Some of the language in the body of the text was inadvertently missed in tracking. Therefore, after round2 edits the final v2 draft of the revised manuscript was carefully read in “tracking” feature in word and “without tracking” in word to ensure that all typos are omitted and that the language is clear throughout the manuscript.

Reviewer 3 Report

Authors have adressed most of my previous comments. However, there are some minor aspects that still have to be taken into account in order to improve the final manuscript.

In the introduction section please the two last objetives still have to be more clearly explained. If  a 20% unblanched peanuts diet was similar to the production performance of hens fed a conventional control diet, what's the point in analysing the effects over quality using another inclusion rate (8%)?. The same for the last paragraph. In addition, this last paragraph is very difficult to understand.

As I commented in my last review, authors should discuss the results of the study in the discussion section taking into account the aims stated in the introduction section. Objetives and the discussion and conclusions should be coherent.

L2005: due to the contents of soybean oils of these diets

L2033: include in the beginning that there are no signifficant differences in quaility syzing...

L2130: clearly influenced by the type the fat source of the diet (soybean oil)

L2161: authors have some hypothetical explanation for these discrepancies, please deepen in this fact....

L2070 the results of yolk color didn't agree with your previous findings with an inclusion of a 20% unblanched high-oleic peanut diet. In L2076-2079 authors commented the influence of the diet. In my oppinion, authors should deepen in explaining the differences in diet ingredients and chemical composition between the two studies that explain the observed discrepancies.

Author Response

Reviewer Comment: Authors have adressed most of my previous comments. However, there are some minor aspects that still have to be taken into account in order to improve the final manuscript.

Reviewer Comment: In the introduction section please the two last objetives still have to be more clearly explained. If  a 20% unblanched peanuts diet was similar to the production performance of hens fed a conventional control diet, what's the point in analysing the effects over quality using another inclusion rate (8%)?. The same for the last paragraph. In addition, this last paragraph is very difficult to understand.

Author response: Thank you for pointing out this in the introduction. We have edited the introduction to more clearly state what our objectives were and the justification for the current study. Please see the revised document. Please see edited section pasted below in red font as reference.

Aka et al., (2020) demonstrated that fermented peanut shells alone could be substituted into feed in place of rice bran at up to 6% inclusion without effecting bird performance, while another study concluded that fermented peanut shell meal could be included up to 5% of the diet to enhance broiler performance [16, 17]. Hence, in this study we aimed to determine the effect of whole-in-shell peanuts and unblanched peanuts on the performance and production of laying hens. Secondly, we aim to determine the effects feeding a whole-in-shell peanut diet or unblanched peanut diet on the quality, lipid and fatty acid content of the eggs produced. In our previous layer feeding trials we demonstrated that layer body weight, and feed intake of hens fed a 24% unblanched peanut diet was similar to that of hens fed a non-conventional control diet containing soy protein isolate, defatted soybean meal and yellow corn [3]. However, hens fed a 24% unblanched peanut diet produced significantly fewer total number of eggs compared to hens fed a control diet containing soy protein isolate [3]. Also, eggs produced from hens fed a 24% unblanched peanut diet were significantly smaller in weight compared to eggs produced from hens fed a control diet containing soy protein isolate [3]. Therefore, in this study we aimed to utilize 1/3 the previous inclusion level of peanuts in the diet (8%) of layers to determine the effects on layer production performance and on egg chemistry and quality. We conjecture that hens fed a 8% unblanched peanut diet will have similar egg production and egg weights to hens fed a conventional layer diet (defatted soybean meal + corn) and similar to hens fed a non-conventional control diet containing soy protein isolate. Additionally, we aim to compare the effects of feeding a non-conventional control diet containing soy protein isolate to the effects of feeding a conventional control diet on layer production performance, egg weights, egg quality and chemistry.

Reviewer Comment: As I commented in my last review, authors should discuss the results of the study in the discussion section taking into account the aims stated in the introduction section. Objetives and the discussion and conclusions should be coherent.

Author response: Per the reviewer’s comments we aimed to improve the clarity of the results/discussion of the findings from each reach objective.

In this study we had 3 objectives that are clearly stated in the introduction of the v2 revised manuscript, 1. to determine the effect of whole-in-shell peanuts and unblanched peanuts on the performance production of laying hens and on egg quality 2. to determine the effects feeding a whole-in-shell peanut diet or unblanched peanut diet on the lipid and fatty acid content of the eggs produced, 3.to compare the effects of feeding a non-conventional control diet containing soy protein isolate to the effects of feeding a conventional control diet on layer production performance, egg weights, egg quality and chemistry.

In the revised manuscript in the body of the text of the results/discussion we discuss the effects of feeding a peanut containing diet on production performance lines 376-388 (objective 1). We discuss the effects of feeding control 1 vs control 2 diet on production performance in the v2 revised manuscript in lines 388-396 (objective 3).

In lines 401-407 of the v2 revised manuscript we discuss the effect of a peanut containing diet (WPN/HOPN) and the control diets (control 1, control 2) on USDA egg grading and sizing (objective 1 and objective 3).

In lines 429-433 of the v2 revised manuscript we discuss the effect of the control diets (control1, control 2) on egg quality (objective 3).

In lines 444-509 of v2 of the revised manuscript we discuss egg chemistry (objective 2).

Reviewer Comment: L2005: due to the contents of soybean oils of these diets

Author response: Per the reviewer’s comment the following statement was added in the results/discussion section lines 496-500 of v2 of the revised manuscript, “). Studies have reported low levels of β-carotene in unrefined extracted soybean oil [Rafalowski et al., 2008]. Nevertheless, these low levels of β-carotene are lost with extraction and refinement of commercially manufacture of soybean oil, utilized in preparation of control-1, WPN and control-2 experimental diets. Extracted peanut oil has been reported to contain small quantities of β-carotene”

Reviewer Comment: L2033: include in the beginning that there are no signifficant differences in quaility syzing...

Author response: Per the reviewers’ comments we open the discussion of the USDA grading and sizing discussion with the following statement in the v2 revised manuscript, “Overall, there were no treatment differences in USDA grading or sizing of eggs between the controls (control-1, control-2) or peanut-containing treatment groups (WPN, HOPN) over the 6-week feeding trial.”

Author response: Per the reviewers’ comments we open the discussion of the egg quality with the following statement, “In general, there were no significant differences in egg quality between the treatment groups, with exception of shell thickness at week 2 and yolk color at week 4 (Table 6).”

Reviewer Comment: L2130: clearly influenced by the type the fat source of the diet (soybean oil)

Author response: We were unable to find line reference# L2130 in the any of the files of the manuscript. However, we edited the statement in v2 of the revised manuscript to more clearly state the dietary factors that have been proven in the literature to affect yolk color in the following sentence, “Feeding trials have demonstrated that egg yolk color has been shown to be greatly influenced by lipid profile (unsaturated vs unsaturated and plant vs animal dietary source of lipid) of the hen diet [33] and the type and concentration of dietary carotenoids with which are transferred along with their pigments to the yolks of the eggs produced[34, 35].”

Reviewer Comment: L2161: authors have some hypothetical explanation for these discrepancies, please deepen in this fact....

Author response:Per the reviewer’s comments we attempt to explain any discrepancies between our previously published peanut feeding trials and the current study in the following ways.

  1. Discrepancy with beta carotene levels between peanut containing treatment groups. Line 499 “Studies have reported low levels of β-carotene in unrefined extracted soybean oil [Rafalowski et al., 2008]. Nevertheless, these low levels of β-carotene are lost with extraction and refinement of commercially manufactured soybean oil, utilized in preparation of control-1, WPN and control-2 experimental diets. Extracted peanut oil has been reported to contain small quantities of β-carotene [41]. Pattee and Purcell, (1967) demonstrated that peanut oil extracted from young peanuts contained 60 µg of β-carotene and 138 µg of lutein per liter, while peanut oil extracted from more mature peanuts had lower concentrations [42]. Peanuts have an indeterminate growth pattern, and thus at harvest peanut pods are collected from a range of differing maturity levels on the peanut plant. Harvest containing a higher percentage of young pods may have elevated levels of β-carotene and lutein in the seed of peanuts. Hence, the whole-in-shell peanuts (WPN) utilized in this study may potentially have had a higher percentage of young peanuts containing elevated levels of β-carotene which enriched the eggs produced in this treatment group, while the unblanched high-oleic peanuts utilized in this study may have had a lower percentage of young peanuts containing β-carotene. In contrast, our previous feeding trials demonstrated that eggs produced from hens fed a HOPN diet had enhanced β-carotene content and yolk color relative to the control eggs [6]. Also, in our previous studies, eggs produced from hens fed a 20% unblanched high-oleic peanut diet had approximately a 1.5-fold increase in β-carotene content relative to the conventional control eggs at week 5 and week 10 of the feeding trial (P<0.0001), that may have been due to the use of younger β-carotene-containing peanut pods relative to the controls.”
  2. Discrepancy with yolk color between treatment groups and previous published study.

While there are some publications demonstrating the enhancement of yolk color from the transfer of carotenoids in the diet to the yolks of the eggs produced and some publications demonstrating that lipids in the diet of egg producing hens enhances yolk color in the eggs produced. There is no data or literature stating the specific mechanisms of action responsible for this in egg-producing hens.  Therefore, we are limited in this discussion of yolk color at this time.

Line 442 of v2 revised manuscript, “Feeding trials have demonstrated that egg yolk color has been shown to be is greatly influenced by lipid profile (unsaturated vs unsaturated and plant vs animal dietary source of lipid) of the hen diet [33] and the type and concentration of dietary carotenoids with which are transferred along with their pigments to the yolks of the eggs produced[34, 35]. Nevertheless, additional research is needed to determine precisely how dietary factors directly influence egg yolk color in layers.”

  1. Discrepancy with egg weights in this study vs previous published feeding trial.

Line 376 we discuss this/these discrepancy in the following statements, “Moreover, there were no significant treatment differences in the average pen body weights (Table 3) over the 6-week feeding trial. This is contrary to our previous layer feeding trials, that showed hens fed a 24% or 20% HOPN diet had reduced body weights and egg weights relative to the study controls [3, 10], suggesting that higher inclusion levels of HOPN may significantly alter body and egg weights in egg-producing layers. In this study, there were no significant differences in average egg weights between the treatment groups over the course of this study, suggesting that inclusion of 8% HOPN or 4% WPN does not alter egg weights relative to control (control-1, control-2) eggs. However, there were significant treatment differences in hen FCR, total dozen eggs produced and feed intake  (Table 4).”

Reviewer Comment: L2070 the results of yolk color didn't agree with your previous findings with an inclusion of a 20% unblanched high-oleic peanut diet. In L2076-2079 authors commented the influence of the diet. In my oppinion, authors should deepen in explaining the differences in diet ingredients and chemical composition between the two studies that explain the observed discrepancies.

Author response: This was an inadvertent error that has been corrected in v2 revised manuscript. In line 434 of the v2 revised manuscript, we state that “In contrast, to our previous reports demonstrate that eggs produced from hens fed a 20% unblanched high-oleic peanut diet had significantly (P<0.0001) enhanced egg yolk color in comparison to eggs produced from the controls when analyzed weekly in a 10-week feeding trial, with no other effects seen on egg quality.”

Also line 504-509 of v2 revised manuscript states that “In contrast, our previous feeding trials demonstrated that eggs produced from hens fed a HOPN diet had enhanced β-carotene content and yolk color relative to the control eggs [6]. Also, in our previous studies, eggs produced from hens fed a 20% unblanched high-oleic peanut diet had approximately a 1.5-fold increase in β-carotene content relative to the conventional control eggs at week 5 and week 10 of the feeding trial (P<0.0001).”

Author response: While there are some publications demonstrating the enhancement of yolk color from the transfer of carotenoids in the diet to the yolks of the eggs produced and some publications demonstrating that lipids in the diet of egg producing hens enhances yolk color in the eggs produced. There is no data or literature stating the specific mechanisms of action responsible for this in egg-producing hens.  Therefore, we are limited in this discussion and more research must be conducted experimentally to broaden our understanding of the mechanisms behind this observation and egg quality assessment. We aim to investigate this further in the future. Any statements that we could make at this moment would be speculative only and not factual as we have not been able to find more information in the published literature.

In lines 440-443 of the v2 revised manuscript we state related comments.
